# Prediction of Acoustic Residual Inhibition of Tinnitus Using a Brain-Inspired Spiking Neural Network Model

**DOI:** 10.3390/brainsci11010052

**Published:** 2021-01-05

**Authors:** Philip J. Sanders, Zohreh G. Doborjeh, Maryam G. Doborjeh, Nikola K. Kasabov, Grant D. Searchfield

**Affiliations:** 1Section of Audiology, The University of Auckland, Auckland 1023, New Zealand; philip.sanders@auckland.ac.nz (P.J.S.); zohreh.doborjeh@auckland.ac.nz (Z.G.D.); 2Eisdell Moore Centre, Auckland 1023, New Zealand; 3Centre for Brain Research, The University of Auckland, Auckland 1023, New Zealand; 4Information Technology and Software Engineering Department, Auckland University of Technology, Auckland 1010, New Zealand; maryam.gholami.doborjeh@aut.ac.nz; 5School of Engineering, Computer and Mathematical Sciences, Auckland University of Technology, Auckland 1010, New Zealand; nkasabov@aut.ac.nz; 6Intelligent Systems Research Centre, Ulster University, Derry/Londonderry BT48 7JL, UK; 7Auckland Bioengineering Institute, The University of Auckland, Auckland 1010, New Zealand

**Keywords:** residual inhibition, amplitude modulated, tinnitus, spiking neural network, prediction, individualised treatment

## Abstract

Auditory Residual Inhibition (ARI) is a temporary suppression of tinnitus that occurs in some people following the presentation of masking sounds. Differences in neural response to ARI stimuli may enable classification of tinnitus and a tailored approach to intervention in the future. In an exploratory study, we investigated the use of a brain-inspired artificial neural network to examine the effects of ARI on electroencephalographic function, as well as the predictive ability of the model. Ten tinnitus patients underwent two auditory stimulation conditions (constant and amplitude modulated broadband noise) at two time points and were then characterised as responders or non-responders, based on whether they experienced ARI or not. Using a spiking neural network model, we evaluated concurrent neural patterns generated across space and time from features of electroencephalographic data, capturing the neural dynamic changes before and after stimulation. Results indicated that the model may be used to predict the effect of auditory stimulation on tinnitus on an individual basis. This approach may aid in the development of predictive models for treatment selection.

## 1. Introduction

Tinnitus is the perception of a sound when no physical external sound source is present, and is often described as a ringing, buzzing, or hissing in the ears [1]. For some, tinnitus can have serious negative impacts on quality of life including disrupted sleep [2], and increased stress levels [3]. Most commonly, tinnitus emerges after cochlear hair cell or neural damage that leads to maladaptive changes in the auditory pathway and brain [4,5]. Many different mechanisms have been proposed to explain the role of the brain in tinnitus perception [6]. One prominent model proposes that decreased sensory input from the periphery leads to a compensatory increase in central gain that amplifies spontaneous activity beyond a threshold or lowers the threshold required to reach conscious perception, and propagates it through the auditory system via increased temporal synchrony within and between regions [4,7]. This model suggests that the tinnitus percept is similar to conscious perception of auditory stimuli in that it relies on a distributed neural network. Evidence suggests that the ‘tinnitus network’ is comprised of auditory and non-auditory regions in the brain [8,9,10]. Various studies have found abnormal connectivity between regions including those involved in perception (auditory, somatosensory, and visual cortices), executive control (frontal and pre-frontal cortices), attention and association (parietal cortex), emotion (limbic areas and insula), and memory (parahippocampal areas) [8,9,10,11,12,13,14].

A better understanding of abnormal activity in these networks is required if targeted tinnitus treatments are to be developed. Currently, there is no cure for tinnitus. If different neurobiological types of tinnitus could be identified, a more tailored approach to intervention may result. Because of the underlying heterogeneity of the aetiology and pathophysiology of tinnitus, it is possible that various treatment options may result in different degrees of improvement, depending on tinnitus subtype. Exploration of different interventions that target these types is needed and could lead to personalised tinnitus treatments.

Temporary suppression of tinnitus following offset of some form of stimulation, e.g., transcranial magnetic stimulation, is termed “Residual Inhibition” [15,16]. Auditory Residual Inhibition (ARI) describes a temporary suppression of tinnitus (partial or complete) after the offset of a masking sound stimulus [17]. Spontaneous activity throughout the central auditory system is suppressed after the presentation of sound in people without tinnitus, and it has been proposed that ARI of tinnitus occurs (at least in part) due to this same phenomenon (for a review see [7]). Because the ARI effect is fleeting, it is not a treatment, but rather a useful technique that can be implemented in tinnitus research. ARI allows for within-subject measures to be taken when a subject is experiencing tinnitus, versus when tinnitus is suppressed, under otherwise identical conditions, as opposed to between-subjects designs, or masking paradigms where sound conditions inevitably must differ between measures. The possibility of within-subject comparisons may also be useful in the development of individualised treatments for this heterogeneous condition. ARI is one of only a few interventions that can temporarily suppress tinnitus in a high proportion of people. Consequently, it is a useful tool for probing tinnitus mechanisms. For these reasons, ARI shows promise as a suitable technique for examining resting-state neural networks that underlie the tinnitus percept [18,19]. A recent study by our group [20] identified ARI in 17 of 30 participants, and found a significant increase in the power spectral density of alpha plus gamma bands of the electroencephalogram (EEG), consistent with a theory that increases in alpha activity represent inhibitory control of synchronised spontaneous activity [19].

Recent research has suggested that amplitude modulated (AM) sounds may be more effective for inducing tinnitus suppression and ARI than constant sounds [21,22,23,24]. AM stimuli have been proposed to normalise tinnitus-related oscillations and hyperactivity in the central auditory system through entrainment [22,23,24]. There is also evidence to suggest that the presentation of modulated auditory stimuli can induce measurable changes in auditory cortex activity that may reflect neuroplasticity [25,26].

Assessment of both EEG and behavioural measures may enable greater precision in treatment selection for given individuals. EEG measures the electrical activity of the brain with excellent temporal resolution across the scalp and is relatively inexpensive. However, maximising the utility of EEG data requires the integration of both temporal and spatial characteristics. Most of the extant analytical techniques create models by separately processing the spatial and temporal information [27,28,29]. They also lack biological plausibility and can be difficult to interpret [30].

In light of this knowledge, the current research applies a novel computational framework based on one of the most promising trends of Artificial Neural Networks: Brain-inspired Spiking Neural Networks (SNN). SNN models have been developed as a neurobiologically-plausible computational architecture that incorporates both spatial and temporal characteristics of data into one unifying model [31]. They are considered a suitable tool for the analysis of Spatiotemporal Brain Data (STBD), where both space and time components are crucial to be learnt.

This research addresses the following aims:To create computational models from EEG data based on brain inspired SNN architecture to explore modelling of neural networks underlying the tinnitus percept and examine how these altered when tinnitus was suppressed in an ARI paradigm.To recognise the patterns of changes in STBD, measured before and after AM and constant treatment across participants.To examine whether AM white noise would produce greater ARI than constant white noise, and whether differences could be captured in the SNN model networks.To assess whether the SNN model could predict which participants experienced ARI and which did not, using baseline data.

## 2. Materials and Methods

### 2.1. Ethical Standards

All subjects gave their written informed consent for inclusion before they participated in the study. The study was conducted in accordance with the Declaration of Helsinki, and the protocol was approved by the University of Auckland Human Participants Ethics Committee (023160).

### 2.2. Participants

Participants were recruited through an advertisement at the University of Auckland Hearing and Tinnitus Clinic. To be included in this study, individuals had to have chronic tinnitus (more than 6 months) and be over 18 years of age. Ten participants (5 males and 5 females, mean age of 57.4 years, range of 22–75 years [S.D. 19.6]) were included in the study.

### 2.3. Materials and Apparatus

#### 2.3.1. Audiometry

The extent of hearing loss was measured for each participant using pure tone audiometry (0.25–16 kHz) with AVANT Audiometry software and a two-channel audiometer (AVANT Stealth, MedRx, Largo, FL, USA). Thresholds were obtained using headphones (DD450, RadioEar, Middelfart, Dk). The modified Hughson-Westlake procedure was employed [32].

#### 2.3.2. Tinnitus Characterisation

Tinnitus characteristics were assessed using Tinnometer software (MedRx, Largo, FL, USA). A two-alternative forced choice procedure was used to match tinnitus pitch; the participant chose which of two tones presented via the audiometer sounded more similar to their tinnitus until a match was reached. Stimuli were presented bilaterally. A pitch match was accepted once the same frequency was selected twice in succession. Tinnitus loudness matches were obtained using the pitch matched frequency.

#### 2.3.3. EEG Acquisition

EEG was recorded in an electrically shielded and sound treated booth (ISO 8253–1:2010) from sixty-four BioSemi active Ag/AgCl recording electrodes. Electrode locations corresponded to the extended international 10/20 system. Electrodes were attached to a fitted BioSemi head cap. Parker Signa gel was applied at each electrode site to ensure reliable conductivity between electrode and scalp. Continuous EEG signals were recorded on a Dell Optiplex 7040 desktop computer at 8192 Hz sample rate with a sixty-four channel BioSemi ActiveTwo system (www.biosemi.com) referenced to the common mode sense active electrode and grounded to the driven right leg passive electrode, and were stored for offline analysis. EEG recordings were made over two ten-minute periods as described below.

#### 2.3.4. Stimulus Presentation

Stimulus presentation was controlled using presentation software (www.neurobs.com) running on a Dell Precision T3610 desktop computer. Written instructions, and loudness scales were presented on a 20 inch Dell LCD monitor in a darkened room. Auditory stimuli were presented through ER-2, 10 Ω insert earphones (Etymotic research). The signal was amplified using a System 3 SA1 Stereo Power Amplifier (Tucker-Davis Technologies). The constant auditory stimulus was broadband white noise with a duration of 1 min, generated using Adobe Audition CC 2014 software. The AM auditory stimulus was created by generating a 10 Hz sinewave pure tone as a carrier wave and multiplying this with the constant stimulus (message wave), before adding the carrier wave. For the ARI paradigm both stimulus types were presented approximately 10 dBA above each participant’s empirically measured Minimum Masking Level (MML). The maximum presentation level for this study was 91 dBA, considered a safe listening level for up to two hours [33]. For the two participants who reported an MML above 81 dBA, the maximum presentation level was applied for the ARI paradigm. The procedure for assessing MML for each stimulus is described below.

#### 2.3.5. Questionnaires

A case history was obtained from each participant [34] and participants completed the following questionnaires: Tinnitus Functional Index (TFI) [35], Depression, Anxiety and Stress Scale (DASS) [36], and Positive and Negative Affect Schedule (PANAS) [37]. 6 tinnitus severity rating scales were used: overall (not a problem 1–5 a very big problem), strength/loudness (not at all 1–10 extremely), discomfort (not at all 1–10 extremely uncomfortable), annoyance (not at all 1–10 extremely), ability to ignore (easy 1–10 impossible), and unpleasantness (not at all 1–10 extremely).

#### 2.3.6. Procedure

Questionnaires were completed and participants were then seated in a comfortable armchair for the remainder of the experiment. Each participant underwent pure-tone audiometry [32], and tinnitus pitch and loudness assessments.

To find the MML for each stimulus type (constant and AM white noise), participants were instructed to tap the up key on the keyboard to raise the volume of the sound (or the down key to decrease the volume) until it just covered their tinnitus, and then press the spacebar to indicate that the MML had been reached. The average of two of these measures was taken as the MML for each stimulus type respectively.

Participants were then fitted with an EEG cap, and electrodes were attached using standard procedures (described above). EEG was recorded over two 10-min periods (Figure 1a–c): 5 min of silence (baseline), 1 min of auditory stimulation, and a further 4 min of silence (ARI assessment). A 10-min washout period between the recording periods was included to ensure that ARI effects had dissipated before the start of the second period (i.e., tinnitus volume had returned to baseline levels). The order of stimulus presentation was counterbalanced so that five participants received the constant stimulus in the first period, and the AM stimulus in the second period, and five received the stimuli in the reverse order. Subjective tinnitus loudness was assessed at the start of the baseline block, and every minute thereafter during the recording periods, with participants rating their perceived tinnitus loudness by pressing a corresponding number between 1 (silent) and 9 (extremely loud) on a computer keyboard.

#### 2.3.7. EEG Data Pre-Processing

Data were down-sampled to 256 Hz and imported into EEGlab [38] for pre-processing. The PREP pipeline [39] was used to re-reference the data to a robust average reference and interpolate bad channels. A 1 Hz high pass Finite Impulse Response filter was applied and ICA (Independent Component Analysis) artefact rejection was applied to remove eye blinks and other ocular artefacts. Each participant’s data were examined to manually select 15 s of continuous data (3840 data points) as close to the post-sound offset response time as possible, and with minimal noise and artefacts (at least five seconds was allowed for response movements to be completed) for analysis. Custom Matlab scripts were used to convert EEGlab datasets to .csv files for input into the SNN model. For each participant, the dataset analysed with SNN model consisted of 20 samples of 192 data points.

### 2.4. Analyses

The current study was organised in a three-phase analysis as follows:Behavioural data analysis based on the scores in the questionnaires (TFI, TSNS, DASS, and PANAS) and the residual inhibition of tinnitus.EEG data were modelled using the SNN architecture to investigate the effects of constant and AM auditory stimulations across participants (responder and non-responder) and to investigate whether the SNN architecture can be used for prediction of response to auditory stimulation.Statistical analysis of the results to evaluate the SNN model significance.

#### 2.4.1. Behavioural Data

To assess ARI for each stimulus type, the tinnitus loudness rating measured at the final baseline point before stimulation occurred was subtracted from the rating measured immediately after the offset of the stimulus.

#### 2.4.2. Computational Modelling of Data in a Brain-Inspired Spiking Neural Network Architecture

The computational modelling, is based on the framework of evolving spiking neural networks, designed to learn from both temporal and spatial information [40]. SNN models are neuro-computational units that are stimulated with respect to neural structure in the brain. The SNN architecture includes several modules: a data encoding procedure; a 3D SNN model that learns from data in an unsupervised mode; a layer of spiking neurons for supervised learning; output classification/regression; optimisation procedure; and finally, interpretation of SNN models and knowledge extraction [40]. For this study, the SNN architecture was designed as the following five steps and shown graphically in Figure 1d–f. These steps are briefly illustrated below:**Data encoding**: Continuous EEG sequences were encoded into discrete spikes, using a threshold-based method where signal increases above a threshold generated a positive spike, and decreases below a threshold generated a negative spike. No spikes were generated if the thresholds were not crossed.**Mapping**: The 3D SNN reservoir was made up of 1471 neurons based on the Talairach brain template [41]. The 64 input neurons (EEG data channels) were positioned in the model according to their Talairach coordinates (*x, y, z*). In the SNN model, after defining a biologically plausible 3D SNN, data were initialised with a Small-World Connectivity rule (SWC) [42] that defines a probability by which a neuron *i* can be linked to a neuron *j* with respect to their internal distance, the greater the distance between *i* and *j* the smaller the connection probability. The generated initial connections are were adapted during the unsupervised learning process which takes into account the temporal dynamics of input data (described in the following section).**Learning**: The model was trained in an unsupervised learning mode, using the Spike Time Dependent Plasticity (STDP) learning rule [43]. SNN models were trained with EEG data before (T1) and after ARI stimulus presentation (T2) in both the AM and constant conditions. The T2 model was subtracted from the T1 model to illustrate differences in connectivity during ARI.**Visualisation**: Visualisations were produced for the T1, T2 and subtraction models in the AM and constant conditions. The numerical information from each trained SNN model was also extracted to evaluate the statistical significance of the models. To this end, for every trained SNN model, an activation level was measured through computing the average value of its connection weights.**Classification**: Finally, the SNN-based methodology was used for prediction of (residual inhibition) response to auditory stimulation in individuals, when only the EEG data from the baseline stage was used. An output layer classifier was trained, in a supervised mode, to learn the association between SNN connectivity at T1 and class label information (responder versus non-responder) determined at T2.

## 3. Results

### 3.1. Participant Characteristics

Duration of tinnitus and psychoacoustic data for the sample are presented in Table 1.

Scores from the TFI, TSNS, DASS, and PANAS questionnaires are presented in Table 2.

### 3.2. Residual Inhibition

Due to the exploratory nature of this work, only a small number of participants were tested, and the behavioural results presented below are descriptive.

Six of the ten participants’ ratings indicated residual inhibition of tinnitus loudness after both constant and AM stimulation compared to the final baseline measurement. They are termed responders in this article (P3, 4, 6, 7, 8, and 10). Of the four non-responders (P1, 2, 5, and 9), two reported no change after stimulation with the constant stimulus, and two reported an increase in tinnitus loudness. After AM stimulation, three reported no change and one reported an increase in tinnitus loudness. Changes in tinnitus loudness ratings between the final baseline measure and immediately after ARI stimulus offset are presented in Table 3.

Figure 2 illustrates the stability of tinnitus loudness ratings for each participant after the silent baseline period versus after auditory stimulation. For most participants, tinnitus loudness remained stable across baseline measurements. A notable exception was participant 5 who described that their severe tinnitus constantly and spontaneously changed in volume, pitch and sound quality. ARI was not observed in this participant. Participant 9 reported a slight increase in tinnitus volume immediately after both types of stimulation but recovered to baseline levels by the next measurement point, a minute later. Loudness ratings were not affected by auditory stimulation for participants 1 and 2. Participants 3, 6, 7, 8, and 10 showed the expected pattern of residual inhibition, where there was an initial suppression of tinnitus loudness, followed by a gradual return to baseline levels over time. Participant 4 described a small ARI effect after auditory stimulation in both conditions, and their loudness ratings then remained stable over the remaining measurement period.

Data from participant 5 were excluded from the EEG/SNN group analyses as they appeared to be an outlier. Their ratings did not follow the same stable pattern as the other non-responders (participants 1, 2 and 9). This participant’s tinnitus was highly variable (i.e., the level and quality of the tinnitus changed spontaneously), and auditory stimulation appeared to worsen it.

### 3.3. Computational Modelling of Data

#### 3.3.1. Visualizations and Mapping

The SNN-based methodology (explained in the methods section above), is constituted of the following steps in this study:Mapping, modelling, classifying and understanding of EEG data.Statistical and quantitative analysis on the SNN models to assess the model significance.

Firstly, a brain-inspired 3D SNN model was designed based on the Talairach brain atlas of 1471 *neurons* (in spiking neural networks, an artificial neuron refers to a computational unit that mimics the behavior of a biological neuron which receives the information, processes it and produces an output). Here, the term *neuron* is used to represent the center co-ordinate of one cubic centimeter area from the 3D Talairach Atlas. The SNN model input neurons are allocated to the 64 EEG channels to transfer their spike trains into the SNN model.

Then, eight separate 3D models were trained with different EEG data sets related to AM (Figure 3) and constant (Figure 4) auditory stimulation across responder group and non-responder groups at baseline (Time 1 [T1]) and post-auditory stimulation (Time 2 [T2]).

As shown in Figure 3 and Figure 4, at baseline (T1) a difference of connection weights between responders and non-responders was evident. In the AM condition, a slight reduction in overall connectivity was observed between T1 (Figure 3a, top) and T2 (Figure 3b, top) for non-responders, and a slight increase in overall connectivity was observed between T1 (Figure 3a, bottom) and T2 (Figure 3b, bottom) for responders.

In the constant condition, a reduction in overall connectivity between T1 (Figure 4a, top) and T2 (Figure 4b, top) was observed for non-responders, and a larger reduction in overall connectivity between T1 (Figure 4a, bottom) and T2 (Figure 4b, bottom) was observed for responders.

To better scrutinize the differences between the SNN models of different stimulations, the connection weights (Wij) of the difference between two correspondingly trained SNN models (T1, T2) were calculated for each group and subtracted (Wij T2−Wij T1). The subtracted connectivity model is depicted in Figure 3c and Figure 4c, which shows the involved model brain areas activated in response to both constant and AM auditory stimulations. Figure 5 shows the distributions of weight values across 64 EEG channels for both responder and non-responder groups before (T1) and after (T2) the AM and constant auditory stimulation.

#### 3.3.2. Statistical Analysis of the SNN Models

The numerical information of the trained SNN models can be analyzed to evaluate the models’ statistical significance. For every trained SNN model, an activation level was measured through computing the average value of its connection weights across EEG channels.

Therefore, for every participant, one SNN model was developed at T1 (baseline) and T2 (after auditory stimulation). The average connection weights for each individual SNN model were calculated as a function of group (responder and non-responder) and time (T1, T2).

The connection weights for each EEG channel were then grouped into five sites for each hemisphere with their topographical features (Figure 6): frontal, temporal, frontocentral, centroparietal and occipitoparietal. It is important to note that these “sites” refer to sections of the modelled data and may not reflect activity of their corresponding anatomical brain regions, since voltages measured on the scalp can have various sources throughout the brain.

A repeated measures ANOVA was performed with Hemisphere (left, right), Site (frontal, temporal, frontocentral, centroparietal, occipitoparietal), Time (T1, T2), and stimulation condition (AM, constant) as within group variables, and Group (responder, non-responder) as a between groups variable. Violations of the assumption of sphericity were corrected using Greenhouse-Geisser corrections. There were significant main effects of *Hemisphere* [F (1,7) = 153.72, *p* > 0.001, *ηp^2^* = 0.96] and *Site* [F (2.24, 15.65) = 224.92, *p* < 0.001, *ηp^2^* = 0.97] and a significant *Hemisphere*Site* interaction [F (4,28) = 6.47, *p* < 0.001, *ηp^2^* = 0.48]. The right hemisphere showed greater mean weights than the left overall and at each site (Occipitoparietal > Centroparietal > Temporal > Frontal > Frontocentral).

Then, separate ANOVAs were conducted for AM and constant stimuli to investigate the effects of auditory stimulation across individuals. This revealed a similar pattern of effects. For AM stimuli, there were significant effects of *Hemisphere* [F (1,7) = 53.24 *p* < 0.001, *ηp^2^* = 0.88] and *Site* [F (4,28) = 123.62, *p* < 0.001, *ηp^2^* = 0.95], and a significant *Hemisphere*Site* interaction [F (4,28) = 5.18, *p* = 0.003, *ηp^2^* = 0.425]. For the constant stimuli, there were significant effects of *Hemisphere* [F (1,7) = 99.51, *p* < 0.001, *ηp^2^* = 0.93] and *Site* [F (1.99,13.96) = 136.42, *p* < 0.001, *ηp^2^* = 0.95].

Separate ANOVAs were then run on data split further by group. For the responder group in the AM condition, there were significant effects of *Hemisphere* [F (1,5) = 53.29, *p* < 0.001, *ηp^2^* = 0.91] and *Site* [F (4,20) = 91.78, *p* < 0.001, *ηp^2^* = 0.95] and a significant *Hemisphere*Site* interaction [F (4,20) = 2.93, *p* = 0.046, *ηp^2^* = 2.37]. For the non-responder group in the AM condition, there was a significant effect of *Site* [F (4,8) = 54.73, *p* <0.001, ηp^2^ = 0.97] and a significant *Hemisphere*Site* interaction [F (4,8) = 4.83, *p* = 0.028, *ηp^2^* = 0.71]. For the responder group in the constant condition, there were significant effects of *Hemisphere* [F (1,5) = 112.18, *p* < 0.001, *ηp^2^* = 0.96] and *Site* [F (4,20) = 102.15, *p* < 0.001, *ηp^2^* = 0.95]. For the non-responder group in the AM condition, there were significant effects of *Hemisphere* [F (1,2) = 20.91, *p* = 0.045, *ηp^2^* = 0.91] and *Site* [F (4,8) = 76.19, *p* < 0.001, *ηp^2^* = 0.97].

No effects of group, time or condition reached significance.

Mean responder (*n* = 6) and non-responder (*n* = 3) group changes were calculated by subtracting the baseline measure from the post-stimulation measure for both the AM and constant stimuli (Figure 7). For the AM stimulus, responder and non-responder groups showed small changes in opposite directions. For the constant stimulus, changes were larger than for the AM stimulus and both groups showed a decrease, with a larger effect for responders.

#### 3.3.3. Individual Differences

At an individual level, weight changes recorded over the temporal region (electrodes F7, F8, FT7, FT8, T7, T8, TP7, and TP8) suggested that responders were more likely to show connection weight changes between baseline and post-stimulation measures than non-responders (Figure 8) in both AM and constant conditions. The direction of change differed between responders: participants 3 and 4 showed a reduction during ARI in both conditions, participants 7 and 10 showed an increase, and participant 8 showed very slight changes in opposite directions between conditions (Figure 8).

#### 3.3.4. Classification and Discrimination

To investigate whether the SNN architecture can be used for prediction of response to stimulation, we trained an SNN model using only the EEG data collected at T1 to predict the output classes at T2. After training the SNN models, a classifier was trained to classify the SNN model activity for the participants’ responses to AM and constant auditory stimulation at T1 (baseline). The predictive outcomes were the two groups of participants: class 1—responder participants to AM; class 2—non-responder participants to AM. The same division of samples for the constant stimulation was defined. A leave-one-out cross validation method was used for the classification experiment.

For the AM stimulus, the SNN was able to classify T1 samples into classes with 97.78% accuracy. It classified responder samples with 98.33% accuracy and non-responder samples with 96.67% accuracy. For the constant stimulus, the SNN was able to classify T1 samples into classes with 93.33% accuracy. It classified responder samples with 98.33% accuracy and non-responder samples with 83.33% accuracy (Table 4).

## 4. Discussion

In this research, brain activity was investigated in relation to ARI in response to two auditory stimulations (constant and AM) in a clinical population, with a view to identifying patterns of modelled brain activity that might be further investigated as predictors of responsiveness and the optimisation of personalised-treatment plans.

An SNN computational model for visualisation, classification and interpretation of the data was applied to two groups of individuals that were characterised as responders or non-responders based on the changes in tinnitus loudness ratings between the final baseline measure and immediately after ARI stimulus offset. We evaluated modelled neural patterns generated across space and time from features of EEG data, capturing the neural dynamic changes associated with before and after auditory stimulation.

### 4.1. Comparison of Stimuli (C vs. AM)

There was no clear advantage of one stimulus type over the other at a group level in either the initial ARI effect on tinnitus loudness ratings or the recovery time; the initial ARI effect was similar for constant and AM stimuli within individuals (Table 3). However, the time it took for ratings to return to baseline after ARI differed between the stimulation conditions for some individuals (Figure 2), suggesting individual variability in the effectiveness of AM versus constant white noise stimuli. The same individuals in our sample responded to both types of stimulus. It may be that differences in the effectiveness of the stimuli on a group level would be clearer in a larger sample, but our results highlight the heterogeneous nature of tinnitus even within a small sample. Heterogeneity is an important factor to consider as it is likely that the development of successful treatments for tinnitus will need to be individualised, or at least targeted at subtypes of the condition [44].

Previous research that has reported superior effects of AM over constant stimuli have tended to use stimuli at or near the tinnitus frequency for each participant [21,22,24], as opposed to broadband noise used in the present study. Another recent study did not find significant differences in tinnitus suppression between tinnitus pitch-matched AM and constant pure tone stimuli, but their results did suggest that AM sounds were better-tolerated by participants than constant sounds [23]. Emotional valence of stimuli was not empirically measured in the present study, but some participants did report a preference for one stimulus type over the other (again there was no obvious winner in this regard), while others reported no preference.

The SNN model shows promise as a method for studying the neural networks underlying tinnitus and ARI in this preliminary research. Visualisations of spiking interactions and connectivity models did suggest differences between ARI responders and non-responders but our samples were small, especially the non-responder group, so strong conclusions about general mechanisms of tinnitus cannot be drawn from our analysis. However, even within this small group, individual differences were apparent in both behavioural and brain-derived data.

Our statistical analyses only detected main effects of hemisphere and site, and therefore we cannot draw conclusions about differences in brain activity between responders and non-responders based on these statistics. The lack of statistical significance could be due to the small number of participants leading to an underpowered analysis, and/or the effect of individual differences within groups “washing out” effects through averaging. The heterogeneous nature of tinnitus means that an individualized approach to tinnitus treatment may be beneficial and our group is concerned with investigating this [45]. Besides increasing the sample size, future studies may also increase sensitivity by focussing on certain frequency bands of the EEG data.

The SNN was used to model the connectivity weights. In the SNN model, connectivity with stronger weights reflects more spike transmission between neurons’ synapses. Based on a Spike Time Dependent Plasticity (STDP) learning rule [43], the more spikes transmitted between two neurons, suggests a stronger connection between that region and other regions. Therefore, the STDP rule captures ‘hidden’ spatiotemporal relations in the STBD stream, in the form of neuronal connections between spatially located neurons in the SNN model. Our investigation of connection weights showed that ARI induced changes could be in opposite directions between participants and even within participants under different conditions. Activity recorded from the temporal electrodes differed between baseline and post-stimulation periods for the majority of ARI responders but little change was observed for the non-responders. Again, these results are indicative of the heterogeneous nature of tinnitus and the need for individualised therapies.

### 4.2. Pattern Classification and Prediction

The SNN model was trained using data collected at baseline to assess whether it could predict which participants would experience ARI and which would not. Despite the lack of group differences in our statistical analyses, the model was sensitive enough to classify data collected at T1 into responder and non-responder groups with high accuracy in both conditions, demonstrating that the SNN approach can be used to predict the effect of stimulation on an individual basis before it is applied. The present results are a promising initial step and a more directed approach, focused on fewer sites and data divided into EEG frequency bands, could yield an improved result. The ability of the SNN model to accurately predict which participants would experience ARI based on their baseline data further demonstrates the merit of the model for tinnitus research and treatment prediction, in agreement with our previous work with masking-sound therapies [45].

This study provides an important initial step towards the utilization of SNN models for tinnitus treatment selection. Studies with larger samples will be required to build and validate truly predictive models. The hope is that individual pre-treatment data from people who suffer from tinnitus can be used to predict whether they will respond to a treatment. This would be an invaluable tool for hearing health clinicians when selecting and creating individual treatment plans for their patients, and may improve patient outcomes.

## 5. Conclusions

### Limitations and Future Research

In the current study, EEG data from a small group were analysed, so the generalisability of the trained models to the wider tinnitus population are yet to be tested. Furthermore, as with other EEG measures, the current scalp recorded data are limited in identifying precise cortical regions generating the activity. Given EEG data reflects activity from the superficial areas of the cortex, a more in-depth investigation of the fundamental brain regions needs to be explored further.

The present study examined modelled networks underlying ARI in response to constant and amplitude modulated white noise stimulation. Future studies could examine whether similar impacts are detected in response to other types of ARI stimuli.

The model offers a potential for researchers to study general changes of neural activity and for predicting possible individual responses to treatment. This work could be developed in the future to have practical clinical applications such as optimal and individualised treatment plans that are tailored specifically to the behavioural data and brain architecture of the individual client. Neurofeedback training has been explored for mitigating tinnitus effects previously and the incorporation of spatiotemporal SNN models of ARI could complement this work [46]. For example, patients could be trained to produce brain activity more similar to that measured during ARI. The likelihood of successful neurofeedback treatments may also be able to be predicted using models based on ARI responder and non-responder classes.

## Figures and Tables

**Figure 1 brainsci-11-00052-f001:**
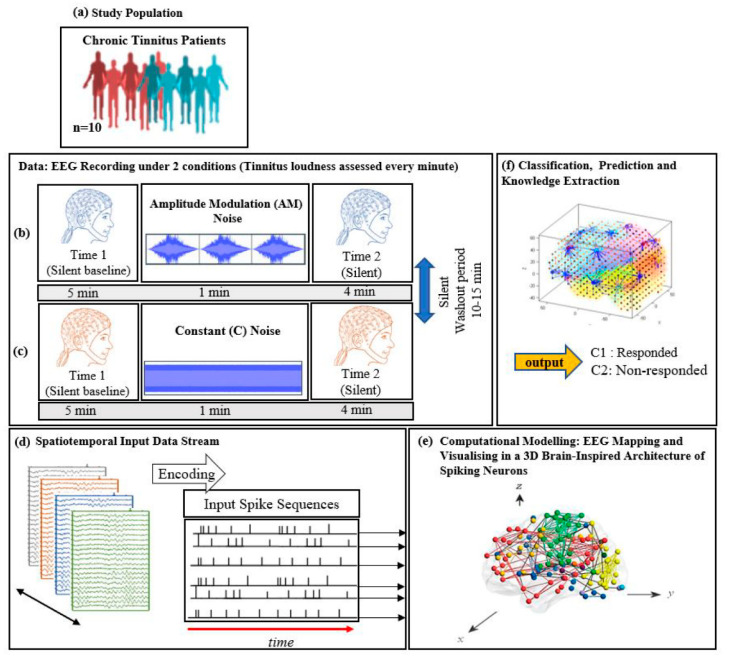
The study protocol: diagram of EEG data collection across (**a**) 10 chronic tinnitus patients that were divided into 2 groups (responder and non-responder based on the changes in tinnitus loudness rating; EEG was recorded over two 10-min periods consisting of a silent baseline period (5 min), a masking stimulus (1 min), and an ARI assessment period (4 min). (**b**) Amplitude modulated and (**c**) Constant white noise were used as auditory masking stimuli, with the order of presentation counterbalanced between participants; (**d**) Illustration of the SNN-based methodology, containing: EEG encoding into spike sequences; (**e**) computational modelling of data into a 3D space of artificial neurons; and (**f**) pattern classification and prediction.

**Figure 2 brainsci-11-00052-f002:**
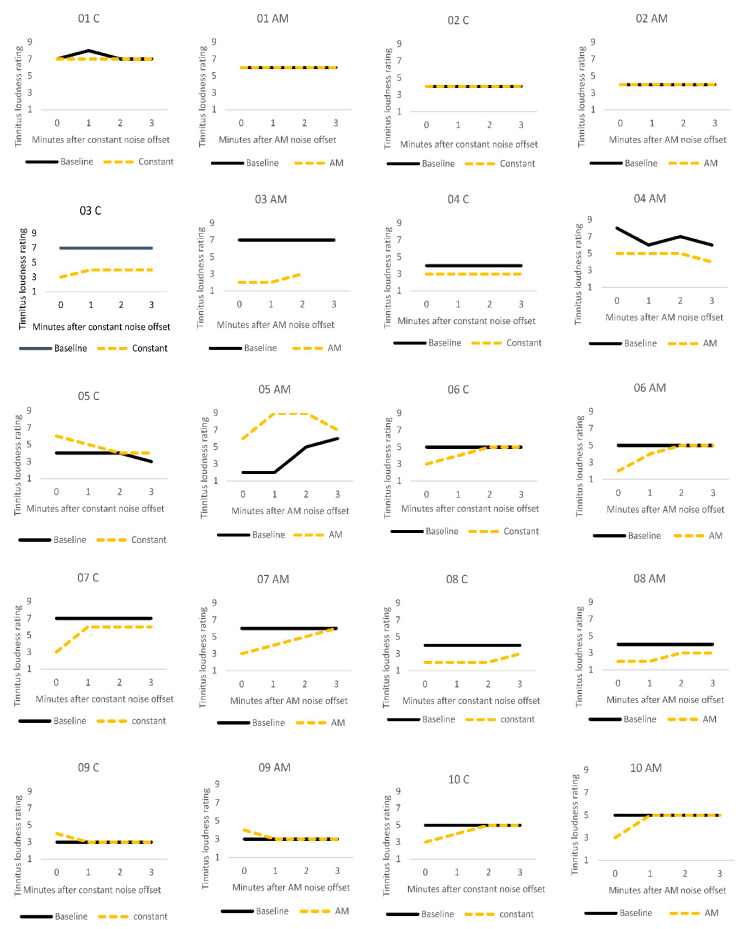
Individual tinnitus loudness ratings for the first three minutes after one minute silent baseline period (solid black line), and after one minute of stimulation (dashed yellow line) with constant (C) or amplitude modulated (AM) white noise.

**Figure 3 brainsci-11-00052-f003:**
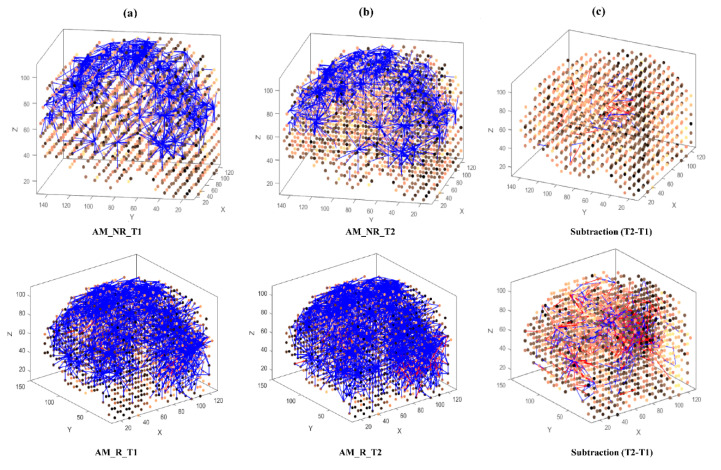
The neuronal connections created for the both non-responder and responder group in the brain-inspired SNN model, reflecting the functional connectivity in response to the amplitude modulated (AM) stimulus condition at (**a**) before (T1); (**b**) after the AM stimulation (T2) and (**c**) differences between the connectivity in the trained SNN models of T1 (before stimulation) and T2 (after stimulation) for each group.

**Figure 4 brainsci-11-00052-f004:**
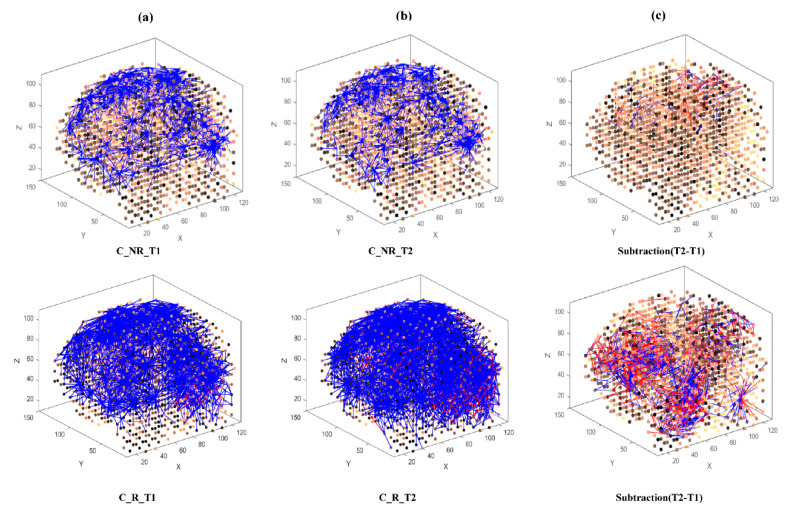
The neuronal connections created for the both non-responder and responder group in the brain-inspired SNN model, reflecting the functional connectivity in response to the Constant (C) stimulus condition at (**a**) before (T1); (**b**) after the C stimulation (T2) and (**c**) differences between the connectivity in the trained SNN models of T1 (before stimulation) and T2 (after stimulation) for each group.

**Figure 5 brainsci-11-00052-f005:**
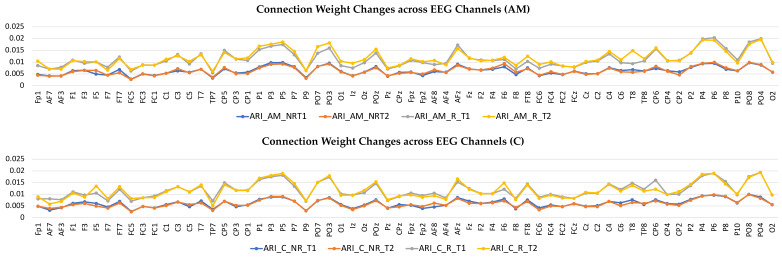
Connection weights changes across 64 EEG channels for both responder and non-responder groups before (T1) and after (T2) the Amplitude Modulation (AM) and Constant (C) auditory stimulation.

**Figure 6 brainsci-11-00052-f006:**
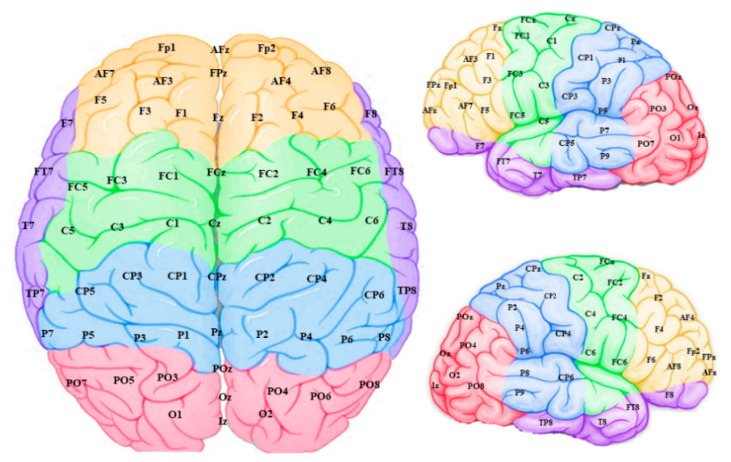
Connection weight data from the model input features (EEG channels) were grouped into frontal (yellow), temporal (purple), frontocentral (green), centroparietal (blue) and occipitoparietal (pink) sites and averaged to scrutinize “regions” within the model.

**Figure 7 brainsci-11-00052-f007:**
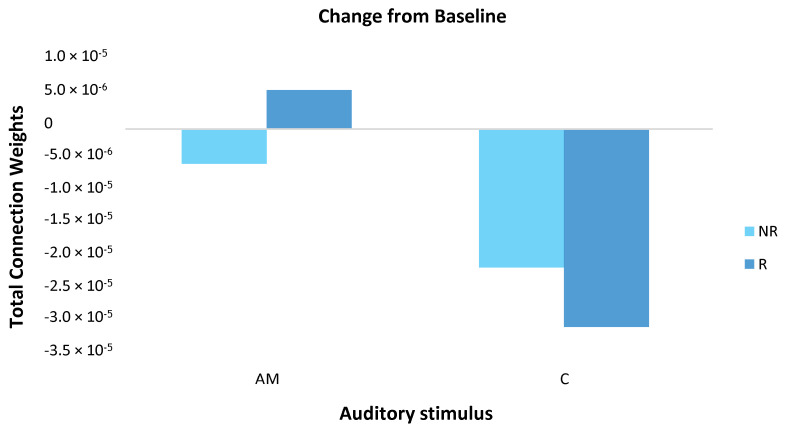
The magnitude of change in total connection weight across the scalp (64 EEG channels) was calculated for responder (R; dark blue) and non-responder (NR; light blue) groups by subtracting the baseline measure from the post-stimulation measure for both the amplitude modulated (AM) and constant (C) stimuli. For the AM stimulus, responders and non-responders showed small in changes in opposite directions. For the constant stimulus, changes were larger than for the AM stimulus and both groups showed a decrease, with a larger effect for responders. Note, these differences did not reach statistical significance.

**Figure 8 brainsci-11-00052-f008:**
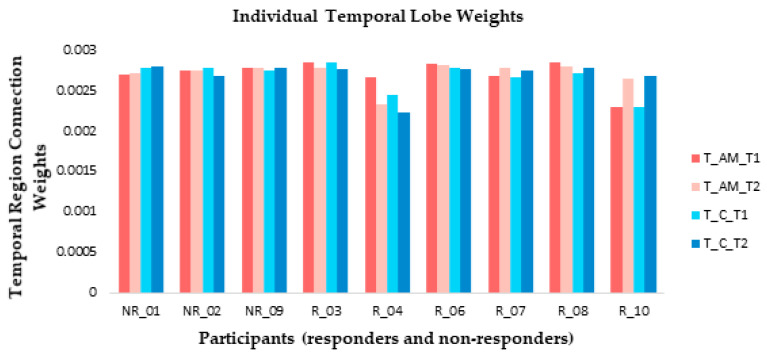
At an individual level, small connection weight changes were observed from temporal lobe electrode recordings between baseline (T1) and ARI (T2) measures for both amplitude modulated (AM) and constant (C) stimuli for the majority of responders (R). Non-responders (NR) generally showed little change between T1 and T2.

**Table 1 brainsci-11-00052-t001:** Individual tinnitus characteristics.

Participant	Duration (yrs)	Pitch (Hz)	Level (dB)	C MML (dB)	AM MML (dB)
1	9	6951	71	37.6	37.7
2	14	2436	70	64.9	59.1
3	9	2420	60	79.9	74.4
4	10	5340	28	58.3	42
5	7	6649	82	67.7	65.9
6	3	5034	80	67.7	65.4
7	13	7151	80	83.1	74.4
8	5	5021	36	36	33.1
9	40	5500	88	73.5	71.7
10	12	1174	68	90.8	90.5
Mean	12.2	4768	66.3	65.9	61.4
SD	10.4	2078	19.9	18	18.5

Yrs = years, Hz = Hertz, dB = decibels.

**Table 2 brainsci-11-00052-t002:** Questionnaire data.

Participant	TFI	TSNS	DASS			PANAS	
	Total	Overall	Depression	Anxiety	Stress	Positive Affect	Negative Affect
1	43.6	3	1	1	3	34	13
2	15.6	3	0	0	1	42	12
3	30.4	3	0	1	1	43	14
4	58.8	2	2	16	15	39	17
5	100	5	37	15	26	31	35
6	41.2	3	11	1	14	31	15
7	26.8	3	2	5	11	40	27
8	13.6	2	0	1	1	20	15
9	12.8	2	0	1	2	50	10
10	23.2	2	0	2	1	44	15
Mean	36.6	2.8	5.3	4.3	7.5	37.4	17.3
SD	26.7	0.9	11.6	6.1	8.6	8.6	7.7

TFI = Tinnitus Functional Index, TSNS = Tinnitus Severity Numeric Scale, DASS = Depression Anxiety Stress Scales, PANAS = Positive and Negative Affect Schedule.

**Table 3 brainsci-11-00052-t003:** Change in tinnitus loudness rating between final baseline measure and ARI stimulus offset.

Participant	Constant	AM	ARI Group
1	0	0	Non-responder
2	0	0	Non-responder
3	−4	−5	Responder
4	−1	−1	Responder
5	3	0	Non-responder
6	−2	−3	Responder
7	−4	−3	Responder
8	−2	−2	Responder
9	1	1	Non-responder
10	−2	−2	Responder

Negative numbers indicate acoustic residual inhibition (ARI), zero indicates no change, and positive numbers indicate increase in tinnitus loudness rating compared to final baseline measurement. AM = amplitude modulated.

**Table 4 brainsci-11-00052-t004:** Classification of 180 EEG samples (20 samples per participant) recorded at T1 into 2 classes: responder at T2 (class1), non-responder at T2 (class2) for amplitude modulated (AM) and constant auditory stimuli. The classification method was leave-one-out-cross validation (LOOCV). The number of correctly classified samples in each class is located in the diagonal of the confusion table.

**SNN-based LOOCV (AM)**				
**EEG data Classes**	**Responder** **Class 1 (predicted)**	**Non-responder** **Class 2 (predicted)**	**Accuracy** **(%)**	**Total Accuracy (%)**
Responder Class 1 (actual)	**118**	2	98.33	97.78
Non-responder Class 2 (actual)	2	**58**	96.67	
**SNN-based LOOCV (Constant)**				
**EEG data Classes**	**Responder** **Class 1 (predicted)**	**Non-responder** **Class 2 (predicted)**	**Accuracy** **(%)**	**Total Accuracy (%)**
Responder Class 1 (actual)	**118**	2	98.33	93.33
Non-responder Class 2 (actual)	10	**50**	83.33	

## Data Availability

The data presented in this study are available on request from the corresponding author. The data are not publicly available due to ethical confidentiality requirements.

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
