# Peer review of "Prediction of Acoustic Residual Inhibition of Tinnitus Using a Brain-Inspired Spiking Neural Network Model"

_brainsci, 2021, doi:10.3390/brainsci11010052_

Round 1
Reviewer 1 Report
Interesting research, well prepared and well presented. Study needs more participants for more accurate results. Would be interesting to recruit more patients in order to reach accurate conclusions. Many thanks.
Author Response
Thank you for taking the time to review our manuscript and for your comments. We agree that a larger study could produce stronger statistical results as pointed out in line 460-462.
However, as we discuss in the manuscript, the heterogeneous nature of tinnitus calls for novel analytical approaches (such as SNN models). Common statistical approaches rely on comparisons of means, this averaging can lead to the loss of important individual differences. As our focus is on producing individualised treatment programs, we were less interested in group average effects and more interested in predictions at the individual level. Therefore, the statistical results took on less importance in our study. However, we wanted to demonstrate that it is possible to extract numerical data for statistical analyses if this important to future research questions.
We agree that research with more participants is needed, but this paper presents an early application of the model to tinnitus. Preliminary investigations such as the present study are useful to explore the power of the technique, and further studies are planned.
The prediction accuracy of the model was impressive, even with the small number of participants. We believe that the model was sufficiently powered because the analysed EEG recording for each participant was divided into 20 samples. Therefore, there were 120 samples (20 for each of the 9 participants). Leave-one-out classification is an iterative process, it trains the model on all samples in the data except one, and then classifies that sample according to the trained model. This process is repeated for the entire dataset (i.e. each sample is left out for training and then classified). This gives us confidence that our classification results are robust and that this method can add to the tinnitus field by advancing prediction of the success of individualised treatment plans.
Reviewer 2 Report
It's very well written and the overall structure is clear and easy to read. The topic is really interesting and well presented. It is clear for either to field's experts and students. Reflections on future research can be improved but the paper submitted could be considered overall satisfactory.
Author Response
Thank you for taking the time to review our manuscript and for your kind comments. If there are more specific suggestions on how to improve the reflections on future research, we will be happy to consider them.
Reviewer 3 Report
The paper is well structured and the idea is original, but needs some changes:
- Exclusion criteria should be included in materials and methods (such as previous ear surgery, meniere syndrome or cochlear hydrops) that could explain the fluctuation of patient number 5.
- Furthermore, it could be better to implement the number of patients to obtain a statistically significant case series.
Author Response
Thank you for taking the time to review our manuscript and for your comments.
Participants were not excluded from participation unless they were under 18 or their tinnitus onset was less than 6 months before participation. The decision to exclude participant 5 from the EEG data analyses was made based on their description of tinnitus (the sounds that they experienced changed regularly), the variability of their baseline and post-stimulation loudness ratings, and because the noise presentation worsened their tinnitus. This was very different to any of the other participants. An explanation is given in line 289-291 of the manuscript.
We agree that a larger study could produce stronger statistical results as pointed out in line 460-462.
However, as we discuss in the manuscript, the heterogeneous nature of tinnitus calls for novel analytical approaches (such as SNN models). Common statistical approaches rely on comparisons of means, this averaging can lead to the loss of important individual differences. As our focus is on producing individualised treatment programs, we were less interested in group average effects and more interested in predictions at the individual level. Therefore, the statistical results took on less importance in our study. However, we wanted to demonstrate that it is possible to extract numerical data for statistical analyses if this important to future research questions.
We agree that research with more participants is needed, but this paper presents an early application of the model to tinnitus. Preliminary investigations such as the present study are useful to explore the power of the technique, and further studies are planned.
The prediction accuracy of the model was impressive, even with the small number of participants. We believe that the model was sufficiently powered because the analysed EEG recording for each participant was divided into 20 samples. Therefore, there were 120 samples (20 for each of the 9 participants). Leave-one-out classification is an iterative process, it trains the model on all samples in the data except one, and then classifies that sample according to the trained model. This process is repeated for the entire dataset (i.e. each sample is left out for training and then classified). This gives us confidence that our classification results are robust and that this method can add to the tinnitus field by advancing prediction of the success of individualised treatment plans.